# Relationship between Rate of Force Development of Tongue Pressure and Physical Performance

**DOI:** 10.3390/jcm11092347

**Published:** 2022-04-22

**Authors:** Syota Saito, Yuta Nakao, Yoko Hasegawa, Koutatsu Nagai, Kyoko Sano, Yuki Uchiyama, Hiromitsu Kishimoto, Ken Shinmura, Kazuhisa Domen

**Affiliations:** 1Department of Physical Medicine and Rehabilitation, Hyogo Medical University Hospital, Nishinomiya 663-8131, Japan; hardler_popo@yahoo.co.jp (S.S.); ynakao33@gmail.com (Y.N.); 2Division of Comprehensive Prosthodontics, Graduate School of Medical and Dental Sciences, Niigata University, Niigata 951-8514, Japan; 3Department of Dentistry and Oral Surgery, Hyogo Medical University, Nishinomiya 663-8131, Japan; kisihiro@hyo-med.ac.jp; 4School of Rehabilitation, Hyogo University of Health Sciences, Kobe 650-8530, Japan; nagai-k@huhs.ac.jp (K.N.); kyokosano@huhs.ac.jp (K.S.); 5Department of Rehabilitation Medicine, Hyogo Medical University, Nishinomiya 663-8131, Japan; yutti@hyo-med.ac.jp (Y.U.); domen@hyo-med.ac.jp (K.D.); 6Department of General Internal Medicine, Hyogo Medical University, Nishinomiya 663-8131, Japan; ke-shimmura@hyo-med.ac.jp

**Keywords:** tongue pressure, rate of force development, aging, frailty, physical performance, maximum tongue pressure

## Abstract

In the assessment of skeletal muscle strength, rate of force development (RFD) is clinically identified as a functional index that reflects the effects of aging, but there are few reports on RFD of the tongue. The purpose of this study was to examine the relationship between RFD of tongue pressure (RFD-TP) and oral and whole-body physical performance in older adults, and to clarify its characteristics. We enrolled adults aged ≥65 years with pathological occlusal contact in premolar and molar regions of teeth in the Tamba-Sasayama area, Japan, from 2017 to 2018. Maximum tongue pressure (MTP) and the speed to reach the maximum tongue pressure (RFD-TP) were evaluated as measures of tongue function. Oral functions related to objective measures of tongue function, such as repetitive saliva swallowing test, oral diadochokinesis, and physical status or performance, such as mini mental state examination, body mass index, skeletal mass index, knee extension force, one-leg standing time, grip strength, walking speed, timed up-and-go test, and five-time chair stand speed was evaluated. No significant correlation was found between MTP and age, but RFD-TP had a significant negative correlation with age. Neither RFD-TP nor MTP showed a significant correlation with oral function. RFD-TP was associated with physical performance, such as knee extension force and one-leg standing time. RFD-TP is more sensitive to aging than MTP. In addition, RFD-TP is related to physical performance and may be useful for the early detection of frailty.

## 1. Introduction

Frailty is a condition wherein there is an increased vulnerability to various stresses due to a decline in physiological reserves associated with aging. It is a preliminary stage leading to the need for nursing care or becoming bedridden. Frailty prevention is important in today’s rapidly aging society because it leads to the maintenance of functional health [1]. Frailty is multifaceted and needs to be approached from various perspectives, such as social, psychological, and physical [2]. Oral frailty is defined as a series of phenomena and processes wherein there are age-associated changes in various oral environments (i.e., oral hygiene) and functions along with a decline in physical and mental reserve capacity that eventually leads to eating dysfunction [3]. It has been reported that there is a twice as high risk for physical frailty, sarcopenia, need for nursing care, and death among people with oral frailty than those without [4]. Therefore, early detection and response is considered a core countermeasure against oral frailty and helps improve healthy life expectancy. Assessing and understanding the oral functions of older adults and screening them for oral frailty is important for maintaining a good quality of life for older adults.

The tongue muscle is composed of rhabdomyosinus muscles, and many studies have evaluated the maximum tongue pressure (MTP)—the maximum muscle force in the anterior lingual location—as a method of assessing muscle strength in oral function [5,6,7]. In addition, MTP is included as an evaluation item in the diagnostic criteria for oral frailty. Low tongue pressure, a state of reduced function of the muscles that move the tongue, may interfere with bolus formation during chewing and spontaneous swallowing and may prevent adequate nourishment [5]. It may also cause dysarthria [6] and impair communication. Therefore, it is important to assess the MTP and prevent low tongue pressure in older adults. MTP has been shown to decrease with age and is associated not only with oral function but also with sarcopenia, jumping ability, grip strength, and other physical performance measures [6,7]. In recent years, the importance of the rate of force development (RFD), which is an index of instantaneous muscular exertion, has been reported in the evaluation of muscular strength of the limbs and trunk, such as knee extensor strength and grip strength, in addition to the maximum muscular strength [8,9,10]. The RFD of knee extensor muscle strength is known to be more strongly related to daily activities than to maximal muscle strength [11]. Furthermore, RFD of knee extensor strength is known to be more susceptible to the effects of aging than maximal knee extensor strength [12,13]. Thus, we hypothesized that, in the tongue muscle, RFD is also more susceptible to aging than to maximal tongue pressure and is related to oral and physical performance.

The RFD of tongue pressure (RFD-TP) can be analyzed using tongue pressure sensor sheets, and RFD-TP can be used to infer swallowing function [14] and articulatory function [15]. RFD-TP is related to swallowing function and sound structure-function. However, to the best of our knowledge, no study has examined the relationship between RFD-TP and aging or physical performance. In addition, it is difficult to use tongue pressure sensor sheets frequently in clinical practice because of the complicated equipment required to prepare for the measurement and the high cost of the disposable sheet. Conversely, the JMS tongue pressure measuring device and the IOPI tongue pressure measuring device is portable, easy to operate, and are widely used in both clinical practice and research. Among these measuring instruments, the JMS tongue pressure measuring instrument was used for RFD analysis using the software provided by JMS in 2017. However, few studies have conducted RFD analysis of tongue pressure, and the direct relationship between tongue pressure and physical performance remains unclear.

Therefore, the purpose of this study was to investigate RFD-TP in older adults using the JMS tongue pressure measuring instrument, clarify the relationship between aging and RFD-TP, and compare RFD-TP to an existing evaluation method, which is the MTP. Furthermore, this study also aimed to examine the relationships among RFD-TP, MTP, and oral and physical performance.

## 2. Materials and Methods

This cross-sectional study was approved by the Institutional Review Board of Hyogo College of Medicine (approval no. Rinhi-0342) and is part of the Frail Elderly in the Sasayama–Tamba Area (FESTA) study. The purpose of the FESTA study was to clarify the associations between lifestyle habits and frailty in older adults, as well as to clarify associations between lifestyle habits and frailty in older adults. Tamba Sasayama is a provincial city with a population of 39,081 (as of the end of December 2021), located in the mountainous region of Japan. Agriculture is the primary industry in this region. It has a noticeably aged population, with adults aged ≥65 years accounting for 31.4% of the population. Written informed consent was collected from participants after discussing the research and registering for inclusion. The data were collected between October 2017 and November 2018.

### 2.1. Participants

Overall, 87 participants (29 men and 58 women, mean ± SE age 74.3 ± 0.7 years) who participated in the FESTA study were included in the investigation of oral function using RFD-TP. The eligibility criteria were as follows: (1) ≥65 years old and with a nursing care requirement of level 1 or less, (2) independence in activities of daily living (ADL), (3) no dysphagia (EAT-10 < 3), and (4) established occlusal contact in premolar and molar regions by the number of remaining teeth, that is, class A in the Eichner classification. The condition of the remaining teeth was confirmed by a dentist. The average number of teeth among participants was 26.8 ± 0.2. Exclusion criteria were as follows: (1) those with a history of cerebrovascular or neuromuscular disease, (2) those with cognitive decline (Mini Mental State Examination (MMSE) score < 20 points), and (3) those with a pacemaker.

### 2.2. Tongue Pressure

Tongue pressure was measured in the sitting position using a JMS tongue pressure measuring instrument (TPM-01, JMS, Hiroshima, Japan). The hard ring at the base of the balloon was fixed with the upper and lower incisors, and the balloon was fixed between the anterior lingual location and the palate of the subject [6]. The balloon was fixed between the anterior lingual location and the palate of the subject. The subject raised the tip of the tongue toward the palate with maximum force. To obtain MTP and RFD-TP, the following instructions were given to the subject: (1) “Please compress the probe with maximum force for 7 s” and (2) “Compress the probe with maximum force as soon as possible”. Each measurement was performed in triplicates. Tongue pressure values were recorded at a sampling frequency of 20 Hz using dedicated software provided by the JMS (http://orarize.com/zetsuatsu/download.php (accessed on 19 April 2022)). All the data obtained were output from the CSV data and used for analysis.

Data analysis, descriptive statistics, and correlations followed those of previous studies. The MTP was defined as the maximum value recorded during the 7 s test duration. RFD was defined as MTP divided by the time it took to reach the MTP (Figure 1).

### 2.3. Oral and Swallowing Function

We examined oral diadochokinesis (ODK) as an oral function measure. The ODK is an alternate motion rate that refers to the rapid repetition of the syllable/ta/within 5 s [6]. The pronunciation of the syllable/ta/involves the use of an anterior lingual location. With the participant in a sitting position, we measured the ODK using the oral cavity function testing device KENKO-KUN (Takei Scientific Instruments, Niigata, Japan) and the number of syllables pronounced. The EAT-10 and repeated saliva swallowing test (RSST) results were examined to infer swallowing function. The EAT-10 is a dysphagia questionnaire consisting of 10 questions [16]. The EAT-10 is used nationally and internationally wherein a score of 3 or more is considered suspicious for dysphagia [17]. The RSST is a screening test for dysphagia and is based on the number of saliva swallows in 30 s. Three or more saliva swallows were considered normal.

### 2.4. Physical Condition and Cognitive Function Assessment

Body mass index (BMI) and skeletal mass index (SMI) were used to assess physical condition. BMI (kg/m^2^) was defined as weight divided by the square of height. Muscle mass of the limbs was measured by the bioelectrical impedance analysis (BIA) method using a body composition analyzer (InBody770, InBody, Tokyo, Japan). The obtained limb muscle mass divided by the square of the height was defined as SMI (kg/m^2^) [18].

In addition, the MMSE was used to assess cognitive function [19]. Patients with a MMSE score of 20 or less were excluded from the study.

### 2.5. Physical Performance

The subjects were instructed to walk at their usual walking pace and the walking speed (m/s) data were collected. Considering the acceleration and deceleration at the beginning and end of walking, the walking section was defined as the distance from 1 m in front of the measurement section (10 m) to 1 m behind the measurement section [20]. The TUG evaluated the time (s) required to sit in the original chair after standing from the sitting position and walking back and forth for 3 m. [21]. Knee extension force was measured using a manual muscle strength meter (Moby, Sakai Medical, Tokyo, Japan). The subject was seated at the end of the table in a 90-degree knee joint flexion position, and the buttocks were not lifted from the treatment table during measurement. The measurement was performed twice with the dominant leg, and the maximum torque value (N) was evaluated [9]. The holding time of the unipedal stance (one-leg standing) was measured from the moment when both hands were placed on the hips and the dominant foot was lifted off the floor, and the time (s) was evaluated until either the hands left the hips, the foot position shifted, or a part of the body other than the supporting foot touched the floor [22]. Five chair stand (5CS) was defined as the time (s) required for the first 5 times of 6 “rise and sit” movements at maximum speed from the chair [23]. A digital grip strength meter (Takei Kikai Kogyo Co., Ltd., Niigata, Japan) was used to measure grip strength. The measurement limb position was standing, and the maximum grip strength of the dominant hand was measured twice each, and the maximum value was used as the grip strength value [24].

### 2.6. Statistical Analyses

The data are shown as median (Q1, first quartile; Q3, third quartile) or mean ± SE. Spearman’s rank correlation coefficient was used to assess the correlation between RFD-TP and MTP with age. To consider the effects of sex in the relationship between tongue pressure and oral/physical performance, the sex differences in each assessment item were examined. Comparisons between the two groups were made after confirming normality and equivariance. Non-normally distributed oral/physical performance data were transformed into square roots for analysis. Two-group comparisons were made using Student’s t-test or the Mann–Whitney U test. Spearman’s rank correlation coefficient was used for correlation analysis. Regression analysis of the association between RFD-TP or MTP and oral/physical performance was performed with age or sex as adjustment variables. Multiple regression analysis (stepwise method, probability of input F ≤ 0.05, probability of removal F ≥ 0.10) was performed with RFD-TP as the objective variable and the factors that showed significant regression coefficients as explanatory variables. All significance levels were set at 5%. Statistical analysis was performed using the IBM SPSS software (version 25.0; IBM Corporation, Armonk, NY, USA).

## 3. Results

### Summary of Subjects

Table 1 shows an overview of the subjects and the correlation between the assessment items and age. As for physical performance, grip strength and knee extension force were significantly higher in males than in females, but there were no significant differences in the other factors. In other words, it is necessary to consider the association with sex differences when examining the relationship between tongue pressure and each factor. In addition, a significant correlation with age was found in physical performance, excluding MMSE, RFD-TP, ODK, and walking speed. This suggests that cognitive function, tongue motor function, and physical performance decrease with age.

Figure 2 shows a scatter plot of tongue pressure and age. The RFD-TP showed a significant negative correlation with age, but the MTP had no significant correlation. Table 2 shows the correlation coefficients between the RFD-TP, MTP, and each variable. RFD-TP showed a significant positive correlation with one-leg standing time and knee extension force. MTP showed a significant negative correlation with walking speed. The results in Table 1 and Table 2 indicated that the evaluation of the association between RFD-TP and MTP, and each oral/physical performance should consider the relationship between aging and sex for each item, respectively.

Table 3 shows the results of the evaluation of the factors involved in RFD-TP and MTP using regression analysis. RFD-TP showed a significant correlation with one-leg standing time or knee extension force. One-leg standing time and knee extension force were not significantly correlated with age and/or sex; that is, faster RFD was associated with longer one-leg standing time and higher knee extension force. Moreover, no item was found to be significantly related to MTP.

Table 4 shows the results of the multiple regression analysis of factors affecting RFD-TP, which showed that RFD was significantly correlated with knee extension force and one-leg standing time.

## 4. Discussion

In this study, we measured tongue pressure in older adults using the JMS tongue pressure measuring instrument and examined the relationship between aging and MTP and RFD-TP. Furthermore, we examined the relationship between RFD-TP and oral and physical performance using age or sex as adjustment variables. The main results of this study are as follows: (1) RFD-TP was affected more by aging than MTP; (2) neither RFD-TP nor MTP was significantly associated with oral function; and (3) RFD-TP was significantly associated with knee extension force and related to physical performance, such as one-leg standing time.

### 4.1. Factors Influencing RFD-TP

This study found no significant correlation between MTP and age. In contrast, the RFD-TP showed a significant negative correlation with age. Suzuki et al. reported that MTP decreases with age in the late-stage elderly, especially in older adults with poor occlusal compensation [25]. However, the present study examined older subjects with good occlusion compensation (subjects classified Eichner A) aged >65 years. Our results showed that RFD-TP decreases with aging even if the occlusion compensation is good. In other words, it was inferred that RFD-TP was more susceptible to age-related functional change than MTP.

This study examined older subjects aged ≥65 years and found no significant correlation between MTP and age. In contrast, RFD-TP showed a significant negative correlation with age. In other words, it was inferred that RFD-TP was more susceptible to aging than MTP.

The anterior lingual location, where tongue pressure measurements were taken, is composed of type II fibers, which are believed to be strongly correlated with tongue movement speed [26]. In addition, RFD has been reported to be a muscle force parameter that reflects changes in instantaneous muscle contraction [8]. Furthermore, Satoh et al. reported that the anterior lingual location is more prone to age-related muscle atrophy than the central part of the tongue and the base of the tongue [27]. It is inferred that RFD in the tongue is strongly affected by aging and is a more sensitive index than MTP in the study of older individuals.

Age-related changes are known to have the following adverse effects on skeletal muscle: muscle atrophy, decrease in the number of muscle fibers [28], decrease in the number of motor units [29], and decreased contraction velocity of the muscle [30]. In addition, Akima et al. recently reported that the amount of fat in skeletal muscle increases with age [31]. Indices of limb skeletal muscle strength, such as knee extension force and grip strength, have been reported to be more susceptible to the effects of aging on RFD than maximal muscle strength [12,13]. The tongue muscle is composed of the rhabdomyosinus muscle and is affected by aging in the same way as skeletal muscle [32]. It has been reported that the tongue undergoes muscle changes in the same way as skeletal muscle, such as age-related muscle atrophy [33], a decrease in the number of muscle fibers [27], and an increase in the amount of fat [34]. The results of this study revealed that RFD is more sensitive to aging than maximal muscle strength in the tongue, similar to knee extension force and grip strength. These results suggest that the tongue muscle undergoes age-related changes similar to skeletal muscle, not only in muscle composition but also in muscle function.

Regarding the effect of sex, both RFD-TP and MTP were not affected by sex in the present study. Previous studies on MTP have shown that there is a sex difference in MTP in the younger group, but the sex difference disappears with older age [35]. In the present study, the MTP was not affected by sex. MTP showed results similar to those obtained previously. RFD-TP, like MTP, showed no sex differences in older adults. Since we have not been able to examine this in the younger group, further studies are warranted.

### 4.2. Relationship between Tongue Pressure and Oral and Physical Performance

Although low tongue pressure in older patients is known to be associated with aspiration and pharyngeal residual [5,36], neither the MTP nor RFD-TP were associated with oral function in the present study. This may be due to the following reasons: (1) the study was conducted in older subjects with preserved swallowing function, and (2) a detailed functional evaluation using video fluoroscopic swallowing study (VFSS) could not be performed because of the effects of radiation exposure. A previous study on RFD of tongue pressure using a tongue pressure sensor sheet compared the findings of VFSS and RFD of tongue pressure in patients with Parkinson’s disease and reported that a decrease in RFD of tongue pressure was associated with a decrease in swallowing function [14]. In this study, tongue pressure was assessed using a JMS tongue pressure measuring device, and it was suggested that RFD-TP is sensitive to aging. Changes in the composition of tongue muscles due to aging are known to affect not only muscle strength but also functional aspects [34,37]. Future studies should compare RFD-TP assessed by the JMS tongue pressure device with a detailed functional assessment using VFSS because the JMS tongue pressure device is simple to operate and easy to apply clinically.

Regarding the relationship between tongue pressure and physical performance, RFD-TP was found to be related to indices of lower limb muscle strength, such as knee extension force and one-leg standing time. Although there is a report on the relationship between appendicular muscle mass and swallowing muscle strength [38], in this study, RFD-TP did not correlate with muscle mass, and the results were related to physical performance.

Age-related muscle atrophy of the lower extremities is twice as large as that in the upper extremities [39,40]. Among the lower extremities, the quadriceps muscle, which is associated with knee extension force, shows accelerated muscle atrophy with age [41]. In addition, knee extension force and one-leg standing are predictors of physical performance decline [42,43]. Regarding the relationship with oral function, a study investigating the relationship between tongue position and isokinetic knee extension exercise reported that when the tongue touches the hard palate, knee flexion peak torque increases [44]. It has been reported that trunk function is involved in one-leg standing balance [45], and that trunk muscle mass is related to swallowing muscles [46]. Thus, knee extension force and one-leg standing balance may be associated with oral function. Here, we found that RFD-TP was associated with knee extension force and one-leg standing time, which tended to reflect aging and decreased physical performance. The association with oral frailty was not clarified in this study because the subjects maintained oral functions such as occlusion and swallowing; however, RFD-TP was more affected by aging than MPT. Since it is also related to physical performance, it may be an evaluation item for the early detection of age-related changes. Since early detection and early response to oral and physical frailty is important for extending healthy life expectancy, further studies on RFD-TP and frailty are warranted.

### 4.3. Study Limitation

The present study had several limitations. First, the tongue pressure was assessed at the anterior lingual location. Second, the study was conducted in older patients with no decline in swallowing function. Therefore, future studies should be conducted in older patients with age-related decline in oral function, such as presbyphagia. We were unable to perform a detailed evaluation of oral functions, such as VFSS. However, as VFSS involves radiation exposure, we did not perform it in this study because we considered it ethically problematic to conduct it in older people without a decline in swallowing function. Finally, as a limitation of the measurement equipment, RFD-TP was recorded every 50 ms, and to evaluate instantaneous movements, the sampling rate was insufficient compared with that in previous studies [14,15]. Therefore, the sampling rate may be inadequate.

## 5. Conclusions

This study investigated the age-related changes in tongue pressure using the JMS tongue pressure measuring device which can analyze the MTP and RFD-TP. The results of our study suggest that the RFD-TP was found to be more sensitive to aging and that the RFD-TP was related to physical performance. Hence, we conclude that the RFD-TP measurement may be useful for the early detection and prevention of frailty at an earlier stage.

## Figures and Tables

**Figure 1 jcm-11-02347-f001:**
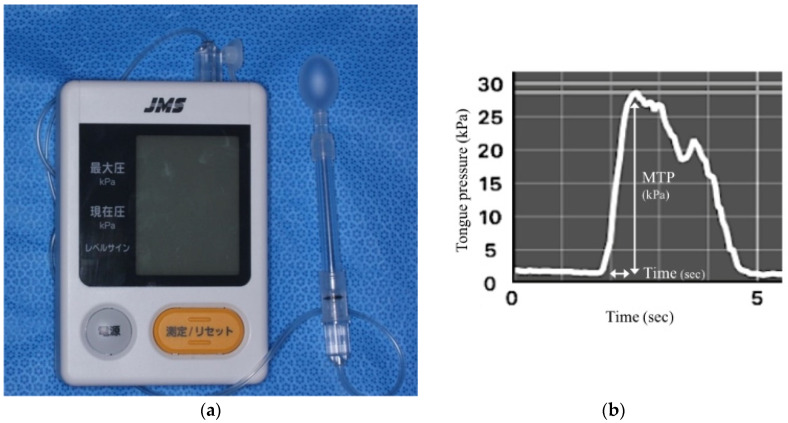
Tongue pressure measurement and assessment. (**a**) Tongue pressure measuring instrument. Tongue pressure was measured using the JMS tongue pressure measuring instrument (TPM-01, JMS, Hiroshima, Japan). The instrument has a power button and a measurement start button, and the LCD monitor displays the maximum tongue pressure and the current pressure value. (**b**) Rate of force development of tongue pressure (RFD-TP). Tongue pressure values were recorded at a sampling frequency of 20 Hz using dedicated software provided by JMS. Instructed the subject to “compress the probe with maximum force as soon as possible”. RFD-TP was defined as maximum tongue pressure (MTP) (kPa) divided by the time (second) it took to reach MTP.

**Figure 2 jcm-11-02347-f002:**
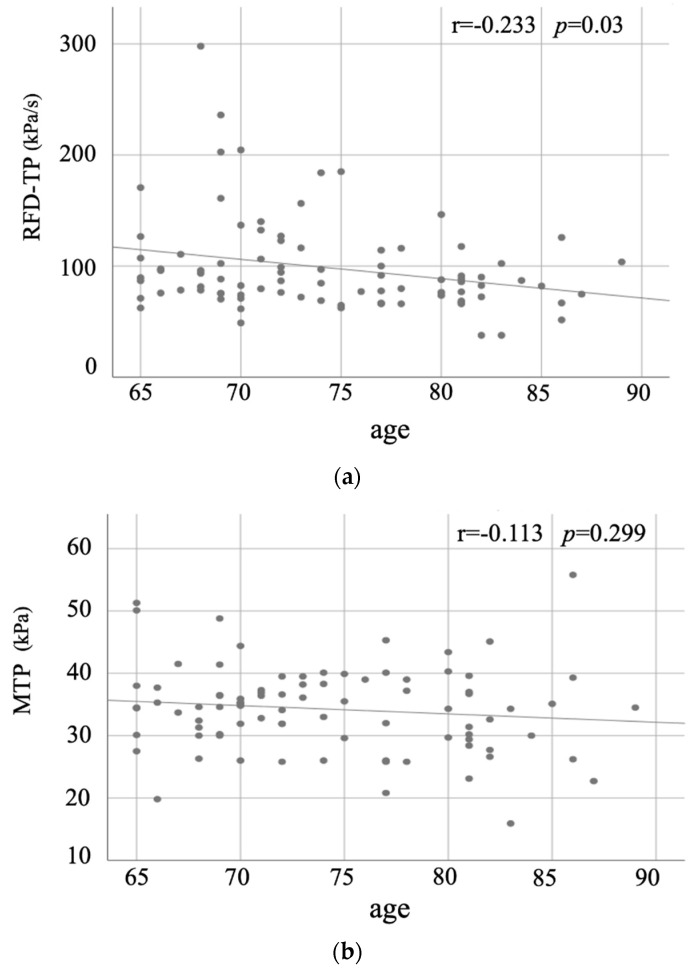
Scatter plots depicting (**a**) the relationship between the rate of force development of tongue pressure (RFD-TP) and age and (**b**) the relationship between the maxi-mum tongue pressure (MTP) and age. The RFD-TP showed a significant negative correlation with age (r = −0.233, *p* = 0.03), but MTP had no significant correlation (r = −0.113, *p* = 0.299).

**Table 1 jcm-11-02347-t001:** Summary of participant characteristics and relationship with aging.

Measurement Variables	Overall (n = 87)	Male (n = 29)	Female (n = 58)	*p*-Value	R^age^	*p*-Value
Physical/Cognitive condition						
Age (years)	74.3 (69.0, 80.0)	75.2 (69.0, 81.0)	73.8 (69.0, 78.0)	0.358	-	-
BMI (kg/m^2^) *	22.6 (20.5, 24.4)	23.8 (22.1, 25.5)	22.0 (20.2, 24.2)	0.001	0.003	0.977
SMI (kg/m^2^) *	6.4 (5.7, 7.2)	7.5 (7.1, 7.9)	5.9 (5.5, 6.2)	<0.001	−0.113	0.299
MMSE (score) #	28.1 (27.0, 30.0)	27.7 (26.0, 30.0)	28.2 (27.0, 30.0)	0.298	−0.295	0.006
Oral function						
RFD-TP (kPa/s) #	98.6 (73.5, 110.6)	109.0(71.6, 151.4)	93.4 (73.9, 102.3)	0.331	−0.233	0.03
MTP (kPa)	34.3 (30.0, 38.2)	33.9 (27.6, 39.4)	34.5 (30.2, 37.8)	0.709	−0.113	0.299
ODK /ta/ (times/5 s) #	31.1 (29.0, 34.0)	30.1 (27.5, 34.5)	31.5 (29.0, 34.0)	0.307	−0.219	0.041
RSST (times/30 s) *	5.1 (3.0, 7.0)	6.0 (4.0, 7.5)	4.7 (3.0, 6.0)	0.012	−0.133	0.221
EAT-10 (score)	0.2 (0, 0)	0.2 (0, 0)	0.2 (0, 0)	0.363	0.126	0.245
Physical performance						
Walking speed (m/s)	1.5 (1.3, 1.7)	1.4 (1.2, 1.5)	1.5 (1.3, 1.7)	0.11	−0.085	0.435
TUG (s) #	6.0 (5.1, 6.5)	5.9 (5.0, 6.3)	6.1 (5.3, 6.5)	0.163	0.581	<0.001
One-leg standing (s) #	32.7 (11.8, 60.0)	34.5 (12.6, 60.0)	31.8 (11.4, 60.0)	0.559	−0.575	<0.001
5CS (s) #	6.87 (5.36, 7.40)	7.61 (5.57, 8.34)	6.49 (5.28, 7.12)	0.192	0.345	0.001
Grip strength (kg) *#	27.6 (22.0, 32.0)	36.4 (30.3, 41.3)	23.2 (20.4, 26.5)	<0.001	−0.295	0.006
Knee extension force (N) *#	358.6 (276.0, 418.0)	467.1 (350.0, 540.5)	304.3 (255.8, 343.0)	<0.001	−0.368	<0.001

Data are shown as median (Q1, first quartile; Q3, third quartile). BMI: body mass index. SMI: skeletal muscle mass index. MMSE: Mini-Mental State Examination. RFD-TP: rate of force development of tongue pressure. MTP: maximum tongue pressure. ODK: oral diadochokinesis. RSST: repetitive saliva swallowing test. TUG: timed up and go test. 5CS: 5-time chair stand test. *p*-value: Comparison between men and women Mann–Whitney U-test *: *p* < 0.05. R^age^: Spearman’s rank correlation coefficient with age, #: *p* < 0.05.

**Table 2 jcm-11-02347-t002:** Correlations between RFD or MTP and evaluated factors.

	RFD-TP		MTP	
	R	*p*-Value	R	*p*-Value
Physical/Cognitive condition				
Age	−0.233	0.03 *	−0.113	0.299
BMI	0.079	0.472	0.184	0.09
SMI	0.173	0.112	0.092	0.401
MMSE	0.21	0.053	0.009	0.937
Oral function				
ODK/ta/	−0.022	0.839	−0.054	0.619
RSST	0.048	0.66	−0.083	0.445
EAT-10	0.021	0.847	−0.035	0.744
Physical performance				
Walking speed	−0.187	0.083	−0.215	0.046 *
TUG	−0.172	0.111	−0.104	0.338
One-leg standing	0.284	0.008 *	0.099	0.361
5CS	−0.177	0.101	−0.175	0.106
Grip strength	0.12	0.268	−0.118	0.277
Knee extension force	0.316	0.003 *	0.029	0.793

RFD-TP: rate of force development of tongue pressure. MTP: maximum tongue pressure. BMI: body mass index. SMI: skeletal muscle mass index. MMSE: Mini-Mental State Examination. ODK: oral diadochokinesis. RSST: repetitive saliva swallowing test. TUG: timed up and go test, 5CS: 5-time chair stand test. R: Spearman’s rank correlation coefficient; *: *p* < 0.05.

**Table 3 jcm-11-02347-t003:** Association of RFD or MTP with oral/physical performance.

RFD-TP	B	β	*p*-Value	β^Age^	*p*-Value	β^Sex^	*p*-Value
Physical/Cognitive condition							
BMI	0.025	0.032	0.782			−0.154	0.187
SMI	2.84	0.266	0.772			0.224	0.772
MMSE	0.171	0.168	0.127	−0.228	0.04		
Oral function							
ODK	0.106	0.011	0.917	−0.271	0.013		
RSST	0.098	0.025	0.823			−0.164	0.145
Physical performance							
TUG	0.34	0.054	0.662	−0.298	0.018		
One-leg standing *	0.249	0.273	0.034	−0.108	0.397		
5CS	−0.047	−0.003	0.956	−0.258	0.023		
Grip strength	0.062	0.012	0.95	−0.275	0.029	−0.19	0.302
Knee extension force *	0.19	0.308	0.048	−0.153	0.214	0.017	0.907
MTP	B	β	*p*-value	β^Age^	*p*-value	β^sex^	*p*-value
Physical/Cognitive condition							
BMI	0.037	0.156	0.181			0.079	0.498
SMI	1.014	0.315	0.105			0.284	0.143
MMSE	−0.01	−0.034	0.764	−0.201	0.079		
Oral function							
ODK	0.381	0.128	0.24	−0.153	0.162		
RSST	1.317	0.093	0.408			0.017	0.877
Physical performance							
TUG	0.271	0.14	0.273	−0.21	0.101		
One-leg standing	0.015	0.051	0.702	−0.106	0.432		
5CS	−0.135	−0.103	0.378	−0.097	0.404		
Grip strength	−0.161	−0.215	0.274	−0.221	0.09	−0.142	0.453
Knee extension force	0.013	0.068	0.683	−0.104	0.431	0.073	0.637

RFD-TP: rate of force development of tongue pressure; MTP: maximum tongue pressure. RFD-TP and MTP were the objective variables (forced imputation method) in simple or multiple regression analysis, and the explanatory variables were the respective oral/physical performance. Age and sex were entered into the equation as an adjustment variable for sex and/or age when a significant correlation was found (Table 1). B: unstandardized coefficient, β: standardized coefficient for each evaluation item. β^Age^: standardized coefficient of age, β^Sex^: standardized coefficient of sex. *: *p* < 0.05.

**Table 4 jcm-11-02347-t004:** Factors associated with RFD-TP.

						Confidence Interval	
Variables	B	S.E.	β	t-Value	*p*-Value	Lower	Upper
One-leg standing	0.100	0.037	0.283	2.714	0.008	0.027	0.173
Knee extension force	0.508	0.206	0.257	2.470	0.016	0.099	0.917
intercept	46.944	13.431		3.495	0.001	20.226	73.662

The multiple regression analysis was performed as follows: Dependent variable: rate of force development of tongue pressure (RFD-TP). Explanatory variables: items significantly associated with RFD-TP in Table 2 and Table 3 (Age, One-leg standing, Knee extension force, 5CS, Grip strength, MMSE and ODK), stepwise selection method (input 0.05, removal 0.10). Statistically significant explanatory variables were shown. B: Unstandardized coefficient, S.E.: Standard error of B, β: Standardized coefficient, Confidence interval: 95% confidence interval for B. Coefficient of determination of the model: 0.190.

## Data Availability

The data of this study are available on request from the corresponding author, Y.H. The data are not publicly available due to restrictions, as they contain information that could compromise the privacy of the research participants.

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
