# Peer review of "Relationship between Rate of Force Development of Tongue Pressure and Physical Performance"

_jcm, 2022, doi:10.3390/jcm11092347_

Round 1

Reviewer 1 Report

This is a good investigation about influence of ageing on oral function.

The description of  the timed up and go test (TUG) was not clear for me (page 4, line 160-161).

Author Response

The description of the timed up and go test (TUG) was not clear for me (page 4, line 160-161).

Author response: Thank you for the comment. We have revised the text to clarify the TUG method as seen on Page 6, line 207-208 of the revised manuscript.

Reviewer 2 Report

This study examines the global physical function predicted by MTP vs. rate of force development-tongue pressure in an elderly population in Japan.  These people had normal swallowing function. One issue that bears further examination is why there was not a trend of decreased MTP by age.  Based on prior data (Suzuki 2020) you should be seeing a decreased MTP with age.  Does this have any implication for the reliability of study measurements?

Author Response

Author response: Thank you for your insightful comment. As you have mentioned, Suzuki's report acknowledged a decrease in MTP with age. The previous study included subjects with Eichner class A, B, and C. Moreover, they also reported that MTP decreased with age, especially in subjects classified as Eichner B and C. However, the subjects in our present study were limited to older adults with better occlusal status classified as Eichner class A, hence, the results of our study differed from the previous study mentioned. As per your suggestion, we added explanations to the “4. Discussion” section of the revised manuscript (Page 8 line 280-285) about the differences between MTP and age from the previous study.

Reviewer 3 Report

This is cross-sectional study about tongue function among unrestricted elderly.  Maximum tongue pressure (MTP) and the speed to reach the maximum tongue pressure (RFD-TP) are selected to represent tongue function. The study is interesting, but I believe there are some points to be adjusted:

  1. In table 1, comparison was made between men and women. However, I do not see any description elsewhere to investigate the difference between  sex.  Please explain why you need to split the patients in two groups depending on sex, and how does it effect your primary outcome.
  2. If your primary outcome is "relationship between aging and RFD-TP / MTP", you should make it clear in your materials and methods section.  
  3. Similarly, making a clear figure to demonstrate the result of your primary outcome will much improve your work.
  4. Please modify your tables to match the "instruction for authors" request.
  5. In page 9, line 289, you mentioned "PD patients". Please avoid unnecessary abbreviation.
  6. In page 10, line 334, you mentioned, "This study investigated the relationship between aging and RFD-TP and between 334 RFD-TP, oral function, and physical performance." I believe you need some language polishing here.

Author Response

In table 1, comparison was made between men and women. However, I do not see any description elsewhere to investigate the difference between sex. Please explain why you need to split the patients in two groups depending on sex, and how does it effect your primary outcome.

Author response: Thank you for the comment. In investigating the relationship between tongue pressure and oral and physical performance, the sex differences in each assessment item were examined in order to consider the effects of sex. As per your suggestion, we added explanations to the “2. Materials and Methods” section of the revised manuscript. (Page 6 line 223-225). Regarding the effect on the primary outcome, as shown in table 3, sex differences did not affect the main outcome. It was described in the “3. Results” section (Page 7 line 262-264), but it was also added to the “4. Discussion” section (Page 8 line 274).

Line 114-122

If your primary outcome is "relationship between aging and RFD-TP / MTP", you should make it clear in your materials and methods section.

Author response: Thank you for pointing it out. As per your suggestion, we have revised the purpose (Page 4 line 131-134) and included explanations on the “Materials and Methods” section in the revised manuscript for clarity. (Page 6 line 222-223)

Similarly, making a clear figure to demonstrate the result of your primary outcome will much improve your work.

Author response: Thank you for your suggestion. We have added a scatter plot to demonstrate the association between age and tongue pressure, which was the primary outcome (Figure 2).  (Page 7 line 253-254).

Please modify your tables to match the "instruction for authors" request.

Author response: Thank you for pointing it out. We have modified the table according to the "instruction for authors" request.

In page 9, line 289, you mentioned "PD patients". Please avoid unnecessary abbreviation.

Author response: Thank you for your keen observation. We have amended this statement as you have pointed out (Page 9 line 347).

In page 10, line 334, you mentioned, "This study investigated the relationship between aging and RFD-TP and between 334 RFD-TP, oral function, and physical performance." I believe you need some language polishing here.

Author response: Thank you for the comment. We have revised the statement accordingly as you have pointed out. (Page 10 line 401-405)

Round 2

Reviewer 2 Report

The explanation regarding the literature on MTP and age is adequate to explain the differences found.  

Author Response

The explanation regarding the literature on MTP and age is adequate to explain the differences found.  

RESPONE: We appreciate your valuable suggestions and comments, which helped us to revise the manuscript to make it better.
Thank you very much.

Reviewer 3 Report

I believe the manuscript is much improved. However, I cannot find the new figure 2 in the revised PDF. Please check. 

Author Response

I believe the manuscript is much improved. However, I cannot find the new figure 2 in the revised PDF. Please check. 

RESPONSE: We wish to express our appreciation to you for the insightful comments, which have helped us significantly improve the paper. Figure 2a and 2b was submitted as a JPEG file, but has been changed to a TIFE file.
It is in a zipped file, so please check it